# Body Image and Quality of Life in Women with Congenital Adrenal Hyperplasia

**DOI:** 10.3390/jcm11154506

**Published:** 2022-08-02

**Authors:** Lea Tschaidse, Marcus Quinkler, Hedi Claahsen-van der Grinten, Anna Nordenström, Aude De Brac de la Perriere, Matthias K. Auer, Nicole Reisch

**Affiliations:** 1Medizinische Klinik und Poliklinik IV, Klinikum der Universität München, LMU München, 80336 Munich, Germany; lea.tschaidse@med.uni-muenchen.de (L.T.); matthias.auer@med.uni-muenchen.de (M.K.A.); 2Endokrinologie Charlottenburg, 10627 Berlin, Germany; marcusquinkler@t-online.de; 3Department of Pediatric Endocrinology, Amalia Children’s Hospital, Radboud Institute for Molecular Life Sciences, Radboud University Medical Center, 6500 HB Nijmegen, The Netherlands; hedi.claahsen@radboudumc.nl; 4Department of Women’s and Children’s Health, Karolinska Institutet, 171 77 Stockholm, Sweden; anna.nordenstrom@ki.se; 5Pediatric Endocrinology Unit, Astrid Lindgren Children’s Hospital, Karolinska University Hospital, 171 77 Stockholm, Sweden; 6Hospices Civils de Lyon, 69002 Lyon, France; aude.brac@chu-lyon.fr

**Keywords:** 21-hydroxylase deficiency, self perception, body appearance, disorder/difference in sexual differentiation, psychosocial determinants

## Abstract

*Objective*: Women with congenital adrenal hyperplasia due to 21-hydroxylase deficiency (CAH) may have poor quality of life (QoL) and low satisfaction with body appearance. We investigated the influence of the patients’ satisfaction with their support on their QoL and body image. *Design*: Retrospective, comparative, Europe-wide study as part of the multicenter dsd-LIFE study. *Methods*: 203 women with CAH were included in this study. We investigated the patients’ QoL and body image compared to a healthy control group. The patients’ satisfaction with their treatment and support in childhood and adolescence as well as in adulthood was assessed by questionnaire and its influence on the patients’ body image and QoL was analyzed by multiple regression models. *Results*: Women with CAH showed worse body image and poorer physical, psychological and social QoL compared to a healthy reference population. The patients’ satisfaction with professional care in the last 12 months was a significant positive predictor for all four domains of QoL (psychological, physical, social, environmental). Dissatisfaction with care in childhood and adolescence and with general support through different stages of life was a significant negative predictor for QoL and body image. *Conclusions*: These results show that women with CAH have poor QoL and body image compared to a healthy reference population. Psychosocial factors such as general and family support, and social interactions with professionals have a substantial impact on QoL and body image in adult females with CAH. This should be taken into account regarding patient care and multimodal therapy.

## 1. Introduction

Congenital adrenal hyperplasia (CAH) due to 21-hydroxylase deficiency is a rare autosomal recessive disorder characterized by impaired cortisol synthesis and in the most severe form also aldosterone synthesis. A further pathognomonic feature is adrenocorticotropic hormone (ACTH)-driven androgen excess due to reduced hypothalamo-pituitary negative feedback [1].

Care of patients with CAH can be burdensome for affected families and the social environment, as well as professional caregivers since patients face lifelong physical and psychosocial challenges due to their conditions [2,3]. In general, CAH is assumed to have greater impact on women compared to men due to unphysiological adrenal androgen excess. During female embryonic development, adrenal androgen excess causes virilization of the external genitalia in the classic form [4] that often requires genital surgery [5]. This may be associated with psychosexual challenges reflected by increased gender-atypical behavior [6], higher bi- or homosexuality [7], less frequent partnerships [8], sexual concerns and impaired satisfaction with their sex lives [9,10], as well as experienced stigmatization in sexual relations regarding their gender as well as their body appearance [11]. 

QoL is a complex concept for which various definitions can be found in the literature [12,13,14,15]. Since in this study QoL was assessed using a World Health Organization (WHO) questionnaire, the following definition should be noted: *“Quality of Life is defined as individuals’ perceptions of their position in life in the context of the culture and value systems in which they live and in relation to their goals, expectations, standards and concerns”* [14]. The WHO concept of QoL includes four domains, which are physical, psychological, environmental and social QoL. This definition thus captures QoL as a subjective evaluation by the individual in relation to internal and external factors. This is in line with other definitions of QoL, where it is described as a multidimensional construct that is influenced by environmental and personal factors and includes both subjective and objective aspects [16].

Looking at previous research, health status [4], genotype [17] and hormonal treatment [18] were found to be important predictors for health-related QoL in CAH and in patients with disorders of sex development (DSD) in general [19]. It was also found that good social support and quality of professional care could be associated with positive outcomes in women with CAH [20]. Findings concerning the QoL in women with CAH, however, are inconsistent [4,8,21,22]. 

Due to adrenal androgen excess and its virilizing effect in female patients, body image is an important outcome measure in women with CAH. In the literature, body image is generally described as a multidimensional construct that contains a person’s attitude towards their own body, including its perception, as well as feelings and thoughts about it [23,24,25,26,27,28]. In this context, the term body image comprises multiple categories, including body satisfaction or dissatisfaction, body acceptance and others [24,29]. Body image is influenced by various aspects, such as cognition, affect and behavior [24,25]. However, studies also show other influencing factors on body image, such as social environment, social media and beauty standards in society [30,31]. Thus, according to research, body image is influenced by both internal and external factors.

When focusing on women with CAH, studies indicate that they are less satisfied with their total body appearance [32], in particular with regard to their subjective view on the physical appearance of their genitals [17]. This seems to be influenced by surgical care, since repeated and multiple genital surgeries lead to dissatisfaction with physical appearance in patients with DSD [33], whereas other studies could not confirm this [22,34]. 

Although there have been studies in other areas on post-operative body image [24,35,36,37,38] some aspects are unique to this study. Women with CAH are typically operated on in childhood which is a vulnerable time [10]. In addition to this, the surgical field is a very private area and repeated examinations are uncomfortable and can lead to feelings of embarrassment among the patients. In addition, sometimes several surgical interventions are necessary [10]. Furthermore, body image in women with CAH is not only influenced by the virilized or operated genitalia, but also by other physical aspects that are caused by hyperandrogenaemia, such as a hirsutism, acne or a deeper voice [39,40]. These symptoms can persist and also affect body image, regardless of surgery.

This is the first study in CAH to investigate if and how professional and social support and quality of professional and psychological care in different stages of life influence body image and different domains of QoL, such as physical, psychological, environmental and social QoL. For this purpose, we firstly compared QoL and body image of our cohort with control groups to identify possible significant differences between these groups. Secondly, we identified important psychosocial factors in patients with CAH. Lastly, we investigated the influence of these psychosocial and other factors on QoL and body image of our patients and identified significant positive and negative predictors for these outcome variables.

## 2. Material and Methods

### 2.1. Study Design

The data from this study were collected within the “dsd-LIFE” study, a European multicenter cross-sectional study funded by the European Commission (7th Framework Program, FP7). From February 2014 to September 2015 participants were recruited in 14 specialized clinical centers in 6 countries (France (*n* = 4), Germany (*n* = 4), Poland (*n* = 2), the Netherlands (*n* = 2), Sweden (*n* = 1) and the United Kingdom (*n* = 1)). Patients were enrolled if they had a medically confirmed clinical or genetic diagnosis of DSD, were at least 16 years of age, and presented to a study center at least once during the course of the study. Patients who did not fulfill the diagnostic criteria or could not consent nor answer the questionnaire by themselves, were excluded. A total of 1040 patients met the inclusion criteria and of these, 203 were females diagnosed with CAH and were therefore included in the study. Ethical approval was received for each study center and all participants gave written informed consent. The study consisted of 2 parts: the first part took place at the respective study center and consisted of a medical interview and an optional medical examination; the second part consisted of a digital patient-reported outcome (PRO) questionnaire and was filled out online. If necessary, a paper pencil form was available. All medical data were pseudonymized and reviewed regarding accuracy of statement. Further information on the theoretical and methodological background of the study can be found elsewhere [41].

Sociodemographic data were collected using the European Social Survey (ESS Round 7, 2015) as well as self-constructed items. The educational status of the participants was measured by the ES-ISCED as a part of the ESS. It categorizes education in “low” (ESISCED 1–2), “middle” (ESISCED 3–5) and “high” (ESISCED 6–7) [18].

To assess QoL the WHOQOL-BREF Questionnaire was used. It consists of 24 items describing 4 different domains of QoL (physical, psychological, environmental, social), with 3 to 8 items per domain. All items are answered on a 5-Point Likert Scale. For each domain 2 different scales can be used with scores either ranging from 0–100 or from 4–20, with higher scores indicating a higher QoL. Both scales can be converted into each other [14].

To measure the participants’ body image, the Body image scale (BIS) was used. The BIS is intended to not only measure the perception about one’s own body but also the feeling about this perception. It consists of questions about 30 body features, which the participants have to rate on a 5-point scale of satisfaction (1 = very satisfied to 5 = very dissatisfied). A total of 27 body features are gender neutral, 3 are gender-related, concerning the internal and external genitals. In this study, the female items were included in the analysis. Eventually, the items’ scores were added up and divided by the number of items, which led to a final mean score ranging from 1 to 5. A higher score indicates higher dissatisfaction with appearance of the own body [27].

Concerning the 4 domains of QoL a reference popualtion was used consisting of 2055 individuals, 927 males and 1128 females [42]. Of these, 250 were young adults aged between 18 and 25 years, and 393 were elderly people who were 66 years and older [42]. For body image, a reference population was used that included 57 female university students aged between 19 and 35 years [36].

As the aim of the study was to assess psychosocial determinants, several domains of the PRO Questionnaire were included, which displayed the participants’ satisfaction with their treatment and support in childhood, adolescence, adulthood resepectively. Therefore, the following domains were included in the analysis: “Diagnosis and Disclosure”, “Your treatment in childhood/adolescence”, “Psychological care and general support in childhood/adolescence”, “Satisfaction with care”, and “Satisfaction with psychological and general support in adulthood”. These categories asked the patients about whether information about the condition, hormone therapy and surgical procedures were given fully and in a sensitive and understandable way during childhood and adolescence. Furthermore, whether the patients were properly informed about treatment options, including risks and side effects, and whether they were satisfied the way physical examinations took place and the doctors’ behavior and knowledge of childhood, adolescence and adulthood. Additionally, it was asked if the patients were satisfied with general support from parents, friends, partners and doctors in childhood, adolescence and adulthood. The items of these categories were either self-constructed or taken from the Child Health Care–Satisfaction, Utilization and Needs (CHC-SUN9) Questionnaire, as well as the Customer Satisfaction Questionnaire (CSQ-4). Most of the items were to be answered on a 4- to 5-point scale. 

### 2.2. Statistical Analysis

In order to compare body image and QoL of women with CAH to a healthy reference population Welch’s tests were conducted as recommended [43]. The Bonferroni–Holm correction was used for multiple testing. 

To assess psychosocial determinants, 38 items were included in the statistical analysis. As these items were not part of a validated instrument but either self-constructed or taken from different measuring instruments, they could not be summed up by predetermined criteria. Therefore, to reduce data for further analysis, an explorative factor analysis was performed. The data’s eligibility for a factor analysis was tested as recommended [44] and based on the results a principal component analysis with oblique promax rotation was carried out extracting 2 factors. In total, the 2 factors explained 45.79% of the variance, factor I with an eigenvalue of 12.04 explained 31.68% and factor II with an eigenvalue of 5.36 explained an additional 14.12%.

To investigate the impact of the psychosocial determinants on the participants’ body images and the different domains of QoL, multiple linear regression models were built for each outcome variable (BIS-Score, physical QoL, psychological QoL, environmental QoL, social QoL). As recent studies showed an influence of age, country and education on patients living with DSD or CAH in particular [19,45], we included these variables in the analysis. Given that previous research has demonstrated that genital surgery has a negative impact on satisfaction of total body appearance in patients with CAH [32] and could influence QoL in patients with DSD [8], it was included in the analysis, as was Body-Mass-Index (BMI) since obesity is a common problem in women with CAH and seems to affect QoL negatively [4]. Ultimately, age at diagnosis was added as this is related to the severity of CAH [10]. The analysis was conducted by using the method of block-wise inclusion.

For the statistical analysis, IBM SPSS Statistics 26.0 was used. Missing values were replaced by the mean value of the respective variable. A significance level of α = 0.05 was determined for the statistical analysis of this study.

## 3. Results

### 3.1. Patient Characteristics

The final study population consisted of 203 female patients with CAH. The largest part of the study population was recruited in Germany (*n* = 86, 42.36%), followed by France (*n* = 52, 25.62%), the Netherlands (*n* = 22, 10.84%), the United Kingdom (*n* = 18, 8.87%), Poland (*n* = 14, 6.90%) and Sweden (*n* = 11, 5.42%). Further details of the study population are displayed in Table 1.

### 3.2. Body Image and QoL in Women with CAH

The basic descriptive statistics of the outcome variables can be seen in Table 2. All four domains of QoL and the BIS-Score were compared to reference populations. While women with CAH showed a significantly poorer body image, physical, psychological and social QoL, they showed significantly better environmental QoL compared to controls. Post-hoc effect size Cohen’s d was analyzed using G*Power, showing small to medium effect sizes [46]. The results can be seen in Table 3.

### 3.3. Correlation between QoL Dimensions and BIS-Score

A correlation matrix for the four domains of QoL and the BIS-Score was built and is presented in Table 4. As BIS-Sore was the only variable that met the criteria for normal distribution Spearman’s rho was used. All four dimensions of QoL intercorrelated significantly (*p* < 0.001) with a positive correlation coefficient ranging from *r* = 0.43 to *r* = 0.67. The BIS-Score correlated significantly (*p* < 0.001) with all four domains of QoL with a negative correlation coefficient ranging from *r* = −0.41 to *r* = −0.56. This indicates that good QoL in one domain is associated with good QoL in the other domains as well as good body image.

### 3.4. Psychosocial Determinants in Women with CAH

The factor analysis revealed two psychosocial factors. Factor I contained all items of the categories “Diagnosis and Disclosure”, “Your treatment in childhood and adolescence”, “Psychological care and general support in childhood and adolescence” included in the analysis and three items of the category “Satisfaction with psychological and general support in adulthood”. These items loading on factor I can be summarized as “Dissatisfaction with care in childhood and adolescence and with general support through different stages of life”.

Factor II contained all items of the category “Satisfaction with care in the last 12 months” and one item of the category “Satisfaction with psychological and general support in adulthood”. The items loading on factor II can be summarized as “Satisfaction with professional care in the last 12 months”.

### 3.5. Influence of Psychosocial Determinants on Body Image and QoL

The results for the multiple regression models with significant predictors for body image (BIS-Score) and the different domains of QoL are depicted in Table 5. Our linear regression models explained 13% of variance of body image as well as 38% (physical QoL), 34% (psychological QoL), 33% (environmental QoL) and 23% (social QoL) of variance of the different domains of QoL. Both psychosocial factors were included in the multiple regression models. Factor I “Dissatisfaction with care in childhood and adolescence and with general support through different stages of life” was a significant positive predictor of the BIS-Score (i.e., lower body image) and a significant negative predictor of all four domains of QoL. Factor II “Satisfaction with professional care in the last 12 months”, was a significant positive predictor of all four domains of QoL but was not a significant predictor of the BIS-Score.

## 4. Discussion

Our results show that women with CAH have worse body image and poorer physical, psychological and social QoL compared to a healthy reference population. In our model, a better body image was predicted by a high level of education, whilst a higher BMI and dissatisfaction with care and general support predicted a worse body image. Furthermore, the study shows that psychosocial factors such as general and family support or social interaction with professionals have a substantial impact on QoL and body image in adult women with CAH. More precisely, it was shown that satisfaction with professional care in the last 12 months has a great impact on women’s QoL, whereas care in childhood and adolescence and social support additionally impact on body image. 

Interestingly, our regression model identified general support and satisfaction with professional care in childhood and adolescence as an influencing factor for body image in women with CAH but not satisfaction with professional care in the last 12 months. These findings are consistent with a study showing that negative experience with parental care during childhood was associated with body insecurity and dissatisfaction with physical appearance later in life in patients with DSD [33]. Positive experience with parental care, on the other hand, was associated with a higher feeling of body attractiveness [33]. It can be speculated that developing a coping ability in childhood and adolescence is an underlying mechanism of these findings. Parental support from health care practitioners may be an important factor resulting in more positive parental care and better body image in their daughters affected by CAH. Additionally, we found that a higher BMI is associated with worse body image. This is consistent with previous studies, both for healthy women [31], and for women with other chronic conditions [47].

Regarding QoL, we found that women with CAH have higher environmental QoL compared to a healthy reference population [42] whilst having lower physical, psychological and social QoL. This is consistent with findings that show decreased QoL in women with DSD, especially in women with CAH, 46,XX and 46,XY- virilized females, using the QoL-AGHDA Questionnaire [8]. It also suggests that a higher degree of virilisation could be associated with a poorer QoL. The UKChase study examining 203 male and female patients with CAH also found decreased QoL across all eight domains of the SF-36 questionnaire [4], as did a study from Norway [48]. However, high doses of glucocorticoids might worsen QoL as well, probably through metabolic changes (e.g., obesity), suppression of the gonadal axis and neuropsychological changes, e.g observed in patients with Cushing’s syndrome. Reduction in high doses of glucocorticoids seems to improve QoL in CAH patients [49].

As the domain social QoL contains aspects of interpersonal relationships and sexual activity [14] our findings are in line with research showing less experience with love and sexual relationships in patients with DSD [50] and decreased satisfaction with sex life in women with CAH [10].

Our regression models identified both psychosocial factors, i.e., satisfaction with professional care in the last 12 months as well as dissatisfaction with care in childhood and adolescence and with general support through different stages of life as significant predictors for the patients’ QoL. Satisfaction with professional care in the last 12 months was found to be a strong positive predictor for all four domains of QoL. Previous research confirms that the degree of satisfaction with professional care is directly associated with health-related QoL in patients with DSD [51]. Dissatisfaction with care in childhood and adolescence and general support at any stage of life, on the other hand, is an important negative predictor for the outcome of women with CAH. We found that it is associated with lower WHOQOL-BREF scores in all four domains, indicating lower QoL. In general, it was shown that children with CAH have an impaired health-related QoL compared to a healthy control group [52], which could negatively influence QoL later in life.

It has to be emphasized that only 17.2% of the participants in our study received psychological support in childhood and adolescence and the same magnitude (18.7%) in adulthood, not allowing further analysis. In itself, however, this is a remarkable and worrying result, as patients with DSD showed a higher frequency of psychological and psychiatric counselling when experiencing severe problems [8]. A study investigating psychological counselling of 110 adults with DSD showed that about one fifth of patients reported that they had never been offered psychological counselling but would have needed it, resulting in the lowest level of satisfaction with care [52]. These findings emphasize the patients’ wishes and need for such care which medical staff should keep in mind while treating these patients throughout their lifetime. 

In our study population, higher educational levels were associated with a better outcome in all four domains of QoL. This is in line with findings regarding physical and health-related QoL in patients with chronic diseases as well as in women with breast cancer [53,54]. In contrast, higher BMI was a negative predictor for physical QoL, which was expected, as obese women in general report lower physical QoL compared to non-obese women [55]. This is of particular interest as obesity is a common and at least in part treatment-related challenge in women with CAH [4,56]. Concerning age at diagnosis, early diagnosis was positively associated with physical QoL. Previous research demonstrated that the timing of diagnosis is associated with the severity of CAH [10] and therefore predicts clinical symptoms [57]. Hence, one explanation for this result could be that CAH forms that are diagnosed at birth are mostly more severe than those diagnosed at infancy and have a greater negative impact on physical health [58]. Lastly, genital surgery was neither a significant predictor for body image nor for the different domains of QoL. This is unexpected as research so far indicates that QoL and satisfaction with body appearance are affected by genital surgery [17]. However, these findings could be explained with the median age of about 30 years of the study sample, which is rather young. As surgical indications and procedures have evolved over the last decades, recent findings indicate that the surgical outcome for female patients with CAH has improved [59]. 

To our knowledge this study is the first investigating the influence of perceived quality of professional and general support on QoL and body image in women with CAH. However, this study also has several limitations. In this work, the four domains of QoL and the body image using BIS score were treated as independent endpoints. Other studies suggest that this could not be the case. A study that examined 581 women with breast cancer using the WHOQOL-BREF and a Body Image Scale for cancer patients [60] showed that the BIS score was a significant predictor for all four domains of QoL in the study population rather than an outcome variable itself [54]. Further research emphasizes these findings as it was shown that body image dissatisfaction is a negative predictor for physical and psychological health related QoL in a healthy study sample [61]. Another limitation is that no standardized instrument was used to assess psychosocial determinants but mainly self-constructed items. Many aspects were subjective and asked retrospectively which could lead to a bias caused by the participants’ current state of mind.

## 5. Conclusions

Our results underline the importance of professional and social support for body image outcomes in this patient cohort. Interestingly, this does not seem to be limited to adult life; perceived support in childhood and adolescence also significantly influences body image later in life. This should be taken into account in the lifelong care of these patients, as their possible body image issues should already be addressed during care in childhood and adolescence. Professional caregivers and the social environment need to be educated in this regard. This is particularly important as the literature has shown the far-reaching effects of positive and negative body images. 

QoL could also be predicted, among other things, by satisfaction with general support and professional care, making it a crucial factor. Therefore, in order to provide good professional support for patients with this rare disease, care should be provided at specialized centers. The goal should be a multimodal therapy concept in which different disciplines cooperate in caring for the patients. This could also address the current issue of clearly insufficient psychological support in patients with CAH that we found in our cohort. There is an urgent need for psychological counselling and therefore support should be offered repeatedly to all patients and families with children with CAH. 

## Figures and Tables

**Table 1 jcm-11-04506-t001:** General characteristics.

		*n*	Total *n*	%
Number of patients			203	
Country			203	
Germany		86		42.36
France		52		25.62
Netherlands		22		10.84
Poland		14		6.90
Sweden		11		5.42
United Kingdom		18		8.87
Education			203	
Low (ESISCED 1–2)		40		19.70
Middle (ESISCED 3–5)		97		47.78
High (ESISCED 6–7)		48		23.65
Unknown		18		8.87
Median age at assessment (min–max) in years	30.28 (15.0–68.0)		203	
Age at diagnosis			203	
Before birth		14		6.90
At birth (0–1 month)		109		53.69
Infancy (1 month–3 years)		33		16.26
Childhood (4–12 years)		15		7.39
Adolescence (13–17 years)		10		4.93
Adulthood (≥18 years)		20		9.85
Unknown		2		0.99
Median BMI at assessment (min–max)	26.45 (15.60–46.40)		199	
Genital surgery			203	
Yes		160		78.82
No		42		20.69
Unknown		1		0.49

Description of the study population: Minimum (min); Maximum (max); Body mass index (BMI).

**Table 2 jcm-11-04506-t002:** Body image and QoL outcomes.

Outcome Variable			*n* = 203		
	M	SD	Range	Min	Max
BIS-Score	2.5	0.7	4.0	1.0	5.0
Physical QoL					
Range 0–100	68.7	18.8	82.1	17.9	100.0
Range 4–20	15.0	3.0	13.1	6.9	20.0
Psychological QoL					
Range 0–100	66.0	18.3	87.5	12.5	100.0
Range 4–20	14.6	2.9	14.0	6.0	20.0
Environmental QoL					
Range 0–100	74.6	15.5	75.0	25.0	100.0
Range 4–20	15.9	2.5	12.0	8.0	20.0
Social QoL					
Range 0–100	64.7	20.8	91.7	8.3	100.0
Range 4–20	14.4	3.3	14.7	5.3	20.0

Description of the outcome variables with mean score (M), standard deviation (SD), range, minimum (Min) and maximum (Max). Body image scale (BIS). The scores for domains of quality of life (QoL) are depicted with range 0–100 as well as 4–20.

**Table 3 jcm-11-04506-t003:** Comparison of outcome variables of women with CAH and healthy reference populations.

Variable	CAH Group			Reference Group			*df*	*t*	*p*	*d*
	*n*	*M*	*SD*	*n*	*M*	*SD*				
BIS-Score	203	2.5	0.7	57	2.4	0.3	196	2.38	0.018	0.19
Physical QoL	203	68.7	18.8	2052	76.9	17.7	239	−5.96	<0.001	0.45
Psych. QoL	203	66.0	18.3	2055	74.0	15.7	232	−6.01	<0.001	0.47
Envir. QoL	203	74.6	15.5	2053	70.4	14.2	237	3.71	<0.001	0.28
Social QoL	203	64.7	20.8	2048	71.8	18.5	235	−4.68	<0.001	0.36

Display of Body image scale (BIS)-Score and the 4 domains of quality of life (QoL) of the study population and reference populations [27,28]. Psych. QoL describes psychological QoL and Envir. QoL describes environmental QoL. Congenital adrenal hyperplasia (CAH). Description of the outcome variables with mean score (M), standard deviation (SD).

**Table 4 jcm-11-04506-t004:** Correlation matrix of the outcome variables.

	BIS-Score	Physical QoL	Psych. QoL	Envir. QoL	Social QoL
BIS-Score					
Physical QoL	−0.42				
Psych. QoL	−0.56	0.67			
Envir. QoL	−0.41	0.62	0.67		
Social QoL	−4.2	0.43	0.60	0.53	

Body image scale (BIS) and quality of life (QoL). Psych. QoL describes psychological QoL and Envir. QoL describes environmental QoL. All correlations were significant with a *p*-value < 0.001.

**Table 5 jcm-11-04506-t005:** Linear Regression models of the outcome variables.

Outcome Variable	Predictor Variable	β	*t*	*p*	*Adj. R* ^2^	*R* ^2^
BIS-Score						
	Country				0.0	0.03
	Sweden	−0.15	−2.04	0.043		
	Educational level				0.01	0.05
	High	−0.18	−2.12	0.036		
	BMI	0.22	3.01	0.003	0.04	0.08
	Psychosocial Factor I	0.24	3.03	0.003	0.12	0.19
Physical QoL						
	Country				0.1	0.13
	France	−0.36	−5.66	<0.001		
	UK	−0.19	−3.15	0.002		
	Educational level				0.17	0.21
	Middle	0.23	3.33	0.001		
	High	0.33	4.54	<0.001		
	BMI	−0.13	−2.19	0.03	0.18	0.21
	Age at diagnosis					
	Infancy	0.2	2.07	0.04	0.22	0.28
	Psychosocial Factor I	−0.27	−3.98	<0.001	0.34	0.39
	Psychosocial Factor II	0.25	3.83	<0.001	0.38	0.43
Psych. QoL						
	Country				0.02	0.05
	France	−0.14	−2.19	0.03		
	Poland	−0.13	−2.13	0.035		
	UK	−0.15	−2.3	0.022		
	Educational level				0.03	0.07
	High	0.15	2.0	0.047		
	Psychosocial Factor I	−0.28	−3.98	<0.001	0.29	0.23
	Psychosocial Factor II	0.38	5.59	<0.001	0.34	0.4
Envir. QoL						
	Country				0.01	0.04
	France	−0.27	−4.16	<0.001		
	Educational level				0.05	0.08
	Middle	0.18	2.5	0.013		
	High	0.22	2.97	0.003		
	Psychosocial Factor I	−0.29	−4.09	<0.001	0.22	0.28
	Psychosocial Factor II	0.38	5.55	<0.001	0.33	0.39
Social QoL						
	Educational level				0.07	0.11
	High	0.28	3.43	0.001		
	Psychosocial Factor I	−0.23	−3.01	0.003	0.16	0.23
	Psychosocial Factor II	0.31	4.17	<0.001	0.23	0.29

Displayed are significant predictors of the outcome variables with β, *t*-value, *p*-value, adjusted *R*^2^, *R*^2^. Body image scale (BIS) and quality of life (QoL). Psych. QoL describes psychological QoL and Envir. QoL describes environmental QoL. Body mass index (BMI).

## Data Availability

The data presented in this study are available on reasonable request from the corresponding author.

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
