# Peer review of "Body Image and Quality of Life in Women with Congenital Adrenal Hyperplasia"

_jcm, 2022, doi:10.3390/jcm11154506_

Round 1

Reviewer 1 Report

Dear authors,

The following comments are intended only to improve the quality of your manuscript, please take the following comments into consideration.

Introduction.

This section provides an extensive justification of the existing problem. But there is no conceptual depth. What is body image? How is it constructed? Is it based on internal or

external elements? and Quality of life?

I suggest the following manuscripts so that you can begin to deal with these concepts:

1. Romano, K. A., Heron, K. E., & Ebener, D. (2021). Associations among weight suppression, self-acceptance, negative body image, and eating disorder behaviors among women with eating disorder symptoms. Women & Health, 61(8), 791-799.

2. Swami, V., Todd, J., Stieger, S., Furnham, A., Horne, G., & Tylka, T. L. (2021). Body acceptance by others: Refinement of the construct, and development and psychometric evaluation of a revised measure–The Body Acceptance by Others Scale-2. Body image, 36, 238-253.

3. Swami, V., Weis, L., Barron, D., & Furnham, A. (2018). Positive body image is positively associated with hedonic (emotional) and eudaimonic (psychological and social) well-being in British adults. The Journal of social psychology, 158(5), 541-552.

4. Wallis, K., Prichard, I., Hart, L., & Yager, Z. (2021). The Body Confident Mums challenge: a feasibility trial and qualitative evaluation of a body acceptance program delivered to mothers using Facebook. BMC public health, 21(1), 1-12.

Previous studies? In the field of bariatric surgery there are quite a few studies that deal with similar aspects to the present one (5,6), what makes it different? Only the population target?

5. Bertoletti, J., Galvis Aparicio, M. J., Bordignon, S., & Trentini, C. M. (2019). Body image and bariatric surgery: a systematic review of literature. Bariatric surgical practice and patient care, 14(2), 81-92.

6. Ivezaj, V., & Grilo, C. M. (2018). The complexity of body image following bariatric surgery: a systematic review of the literature. Obesity reviews, 19(8), 1116-1140.

Method

Participants

What were the inclusion and exclusion criteria for participation in the study?

Results

In order to know whether the results are significant or not it is necessary to know the rate of explanation of the differences between the population add eta squared.

Author Response

jcm- 1799124

Body Image and Quality of Life in Women with Congenital Adrenal Hyperplasia

Dear editors, dear reviewers, 

We appreciate the careful reviewing of our manuscript and are grateful for the reviewers’ comments and suggestions which have helped to improve it.

We addressed the reviewers’ comments in the attached point-to-point reply. We are convinced that we provide detailed responses to all concerns and hope that our manuscript will now be acceptable for publication in the Journal of clinical medicine.

We are looking forward to your feedback.

Yours sincerely,

Nicole Reisch on behalf of all co-authors

Reviewers’ comments to author:

Reviewer 1 

Dear authors,

The following comments are intended only to improve the quality of your manuscript, please take the following comments into consideration.

Introduction: This section provides an extensive justification of the existing problem. But there is no conceptual depth. What is body image? How is it constructed? Is it based on internal or external elements? and Quality of life?

Previous studies? In the field of bariatric surgery there are quite a few studies that deal with similar aspects to the present one (5,6), what makes it different? Only the population target?

Reply: Thank you for these important comments and the suggested literature. We have revised the introduction focusing on the concepts of quality of life and body image. As part of this, we have also decided to change the term body acceptance to body image in the title and throughout the manuscript, as body acceptance is only a part of body image. Since we used the “body image scale” this seems methodically more appropriate.

Furthermore, we also emphasized the differences between post-operative body image in women with CAH and other cohorts. 

Method

Participants: What were the inclusion and exclusion criteria for participation in the study?

Reply: We added the inclusion and exclusion criteria for the study in the method section (lines 107-110).

Results: In order to know whether the results are significant or not it is necessary to know the rate of explanation of the differences between the population add eta squared.

Reply: Thank you very much for the important comment. Since we determined the group differences by using Welch test, we have reported Cohen's d as post-hoc effect size (lines 195/196; Table 3). 

Reviewer 2

Dear Authors.

The following is a review of the article entitled " Body Acceptance and Quality of Life in Women with Congenital Adrenal Hyperplasia" which hypothesis and aims: Women with congenital adrenal hyperplasia due to 21-hydroxylase deficiency (CAH) may have poor quality of life (QoL) and low satisfaction with body appearance. We investigated the influence of the patients’ satisfaction with their support on their QoL and body acceptance. Thank you very much for thinking of me as a reviewer for this study.

After carefully reading the manuscript, I set forth comments and suggestions for the editors:

General considerations: The study is interesting and novel. Adding the lines of each paragraph in this manuscript would have helped the review.

Reply: We are sorry for the inconvenience, we have now added the lines.

Abstract: Correct.

Keywords: Correct.

Introduction: The development of this section is correct, but the objective of the study should be clearly stated at the end of the introduction. It does not appear.

Reply: We thank the reviewer for this comment. We have revised the introduction, also based on the comments of reviewer 1 and now clearly stated the objective of the study at the end of the introduction (lines 94-99).

Materials and Methods: The development of this section is correct. You might add,

What reference population was used to assess body acceptance in women?

Reply: The reference population for body image was a cohort of 57 female university students aged between 19 and 35 years. This information is now given in the methods section of the manuscript as well as the respective reference (lines137/138).

Statistical analysis: Normality and homogeneity tests should be performed on all data obtained.

Reply: We thank the reviewer for this important note. Since we took the reference data from the literature, unfortunately we could not conduct tests for normal distribution and variance homogeneity. For this reason, to avoid bias, we performed the Welch test to compare the study and control groups (line 156-158).

Results: In each table, in notes, the explanation of all acronyms should appear.

Reply: We added an explanation of all acronyms in the notes to the tables.

Discussion: Correct.

Possible limitation: The study should analyze the possible influence of the BMI of each patient on the body acceptance variable.

Reply: BMI in our regression model was indeed identified as a significant negative predictor of body image. This is in line with previous literature. We have added this important aspect in the discussion (lines 267-269).

Conclusions: The "Conclusions" section should be added. Added practical application and limitations.

Reply: Thank you for this important suggestion. We added the “conclusion” section with a brief summary of the results and consequences for practical application.

References: Incorrect 3,5, 14, 20.

Reply: We double checked the references and now consistently use Vancouver style. We also had the manuscript checked for spelling and grammar.

Reviewer 2 Report

Dear Authors.

The following is a review of the article entitled " Body Acceptance and Quality of Life in Women with Congenital Adrenal Hyperplasia" which hypothesis and aims: Women with congenital adrenal hyperplasia due to 21-hydroxylase deficiency (CAH) may have poor quality of life (QoL) and low satisfaction with body appearance. We investigated the influence of the patients’ satisfaction with their support on their QoL and body acceptance. Thank you very much for thinking of me as a reviewer for this study.

After carefully reading the manuscript, I set forth comments and suggestions for the editors:

General considerations: The study is interesting and novel. Adding the lines of each paragraph in this manuscript would have helped the review.

Abstract: Correct.

Keywords: Correct.

Introduction: The development of this section is correct, but the objective of the study should be clearly stated at the end of the introduction. It does not appear

Materials and Methods: The development of this section is correct. You might add,

What reference population was used to assess body acceptance in women?

Statistical analysis: Normality and homogeneity tests should be performed on all data obtained.

Results: In each table, in notes, the explanation of all acronyms should appear.

Discussion: Correct.

Possible limitation: The study should analyze the possible influence of the BMI of each patient on the body acceptance variable.

Conclusions: The "Conclusions" section should be added. Added practical application and limitations.

References: Incorrect 3,5, 14, 20.

Author Response

(The authors gave the same response as above.)

Round 2

Reviewer 1 Report

Great job!